# Is Lower Trust in COVID-19 Regulations Associated with Academic Frustration? A Comparison between Danish and German University Students

**DOI:** 10.3390/ijerph19031748

**Published:** 2022-02-03

**Authors:** Julia Ballmann, Stefanie M. Helmer, Gabriele Berg-Beckhoff, Julie Dalgaard Guldager, Signe Smith Jervelund, Heide Busse, Claudia R. Pischke, Sarah Negash, Claus Wendt, Christiane Stock

**Affiliations:** 1Institute of Health and Nursing Science, Charité–Universitätsmedizin Berlin, Corporate Member of Freie Universität Berlin and Humboldt-Universität zu Berlin, 13353 Berlin, Germany; Julia.ballmann@charite.de (J.B.); stefanie.helmer@charite.de (S.M.H.); 2Unit for Health Promotion Research, University of Southern Denmark, 6705 Esbjerg, Denmark; gbergbeckhoff@health.sdu.dk (G.B.-B.); jguldager@health.sdu.dk (J.D.G.); 3Physiotherapy Department, University College South Denmark, 6705 Esbjerg, Denmark; 4Section for Health Services Research, Department of Public Health, University of Copenhagen, 1014 Copenhagen, Denmark; ssj@sund.ku.dk; 5Department Prevention and Evaluation, Leibniz Institute for Prevention Research and Epidemiology–BIPS, 28359 Bremen, Germany; busse@leibniz-bips.de; 6Institute of Medical Sociology, Centre for Health and Society, Medical Faculty, Heinrich Heine University Duesseldorf, 40225 Duesseldorf, Germany; claudia.pischke@hhu.de; 7Institute for Medical Epidemiology, Biometrics and Informatics (IMEBI), Interdisciplinary Center for Health Sciences, Medical School of the Martin-Luther University Halle-Wittenberg, 06112 Halle (Saale), Germany; sarah.negash@uk-halle.de; 8Department Sociology of Health and Healthcare Systems, University Siegen, 57068 Siegen, Germany; wendt@soziologie.uni-siegen.de

**Keywords:** students, university, COVID-19, frustration, governmental trust, Denmark, Germany

## Abstract

Despite the proximity of both countries, Danes and Germans differ in the level of trust in their government. This may play a role with respect to the disruptive impact of the COVID-19 pandemic on university students. This study investigated the association between trust in governmental regulations, trust in university regulations, risk perceptions, and academic frustration among Danish and German students. As part of the COVID-19 International Student Well-being Study, an online survey was distributed among university students in participating European and non-European universities. In Denmark, 2945 students and Germany, 8725 students responded to the questionnaire between May and July 2020. Students from both countries reported approximately the same level of academic frustration concerning their progress and quality of education. However, German students perceived a higher risk of contracting SARS-CoV-2 compared to Danish respondents. Danish students showed higher trust in their government’s COVID-19 regulations than German students. Lower trust in government and university COVID-19 regulations and higher risk perception were associated with higher academic frustration. These results indicate that the level of trust in COVID-19 regulations might have an impact the overall frustration of students regarding their study conditions.

## 1. Introduction

Students’ experiences during their university years are influential in their later life [1]. COVID-19 regulations impacted these experiences, as a lot of the normal student lifestyle was no longer possible [2]. Most universities in Europe, including in Germany and in Denmark, offered only online teaching during the first lockdown. Online teaching limited regular interaction with peers and lecturers [3]. Many university students suffered from academic frustration because of quick, disorganized switches to online education and examinations [2,4]. Frustration is described as a ‘blocking or prevention of a potentially rewarding or satisfying act or sequence of behavior, or the emotional response to such hindrance’ [5]. Such hindrance occurred when students had to stay at home for online teaching. Negative feelings such as disappointment, worry, fear, anxiety, tension, or anger can be a consequence of this kind of interruption [6].

It remains open, however, whether academic frustration evolved to the same extent in different countries, because the conditions under which students live and are financed differ from country to country across Europe. Our study focused on Germany and Denmark, because the countries are similar with respect to demographic characteristics, extent and date of COVID-19 measures, and geographical and cultural proximity [7], but at the same time there is a difference in financing due to different welfare regimes. While Danish students are not charged any tuition fees and receive a monthly allowance from the government to cover basic living costs [8], German students have to pay an admistrative fee (€200 to €350) and only a small percentage of students receive a monthly allowance from the government (grants and loans system).

Despite the financial security, 47% of Danish students considered themselves stressed during the pandemic [9]. Even before the pandemic, the general competitive environment and the various educational options and breadth of possibilities seemed to cause mental health issues in 18 to 28 year olds from Denmark because of the pressure felt to be responsible for their own success and failure [10]. Like the paid workforce, university students often work under time pressure and with deadlines and demands, which results in a high level of perceived stress [11]. Before the pandemic, a sizable proportion of German students also reported mental health vulnerabilities due to studying [12,13]. It is still not clear from the existing literature how the mental health of German students changed during the pandemic, while most studies indicate a deterioration in mental health for this population. One smaller-scale study indicated no difference in mental health, stress, or depression among students before and during the pandemic in Germany [13]. However, in the same study, the fraction of students who showed vulnerable patterns in 2019, such as burnout, overexertion, or being unambitious, increased during the pandemic. The percentage of students with an unambitious pattern rose from 26% to 31% during the follow up [13]. Another German study, where 10% of the students reported that they could not cope well with the distance learning situation [12], supports the theory that students’ mental health worsened during the pandemic. Similarly, a German study found that social distancing and the loss of institutional in-person teaching led to feelings of isolation and loneliness in students [14]. A cross-sectional analysis of the association between study conditions and depressive symptoms during the pandemic in another German sample of 5021 university students revealed that when students perceived study conditions to be good, they also reported fewer depressive symptoms [15].

Well-being during societal lockdowns may also interfere with the trust in the urgency and adequacy of the regulations set in place. There is evidence for a strong relationship between trust and life satisfaction, and institutional trust was found to be a predictor for general life satisfaction during the first wave of the pandemic in Germany [16]. Political and governmental trust during the pandemic is partly based on perceptions of citizens feeling that authorities interfered enough [17]. Political trust is considered the foundation of the adherence and compliance of people with regulations [18]. A study in Germany looked at how the age, sex, and marital status of a person impacted their level of political trust during the pandemic [19]. This study showed that being married increased political trust but having children in schools decreased trust in women [19]. It was also found that during this time, governments were judged by the timing and appropriateness of their countermeasures against the pandemic [17]. A study in Denmark and Sweden conducted between March and June 2020 showed that Danish respondents trusted their authorities more during the pandemic than Swedish respondents [20]. In addition, a worldwide study supported that Danish people had a higher level of trust in their government with handling the COVID-19 crisis (4.4 on average on a scale from 1 (strongly distrust) to 5 (strongly trust) than Germans (3.6 on the same scale) [17]. In the years before the pandemic, Danish citizens also reported a higher level of political trust than in Germany [21]. The speed of imposing regulations might partially explain this difference. Denmark was the second country in Europe to impose a widespread lockdown announced on 11 March 2020. Comparable measures were introduced on 22 March in Germany. Another difference might be the responsibility for regulations across the country. While Denmark had national regulations, Germany has a federal state structure, and therefore policies differed in the federal states. The extent and dates of these measures and when they were lifted were in general comparable between Germany and Denmark [7]. The measures restricted multiple fundamental civil rights, from enforcing keeping distance, a ban on group assembly, and wearing face masks, to distance learning and working. In Denmark, university classes were switched to being taught online from the beginning of the lockdown (11 March 2020), while the semester had already started on 1 February 2020. In Germany, university classes were taught online from the beginning of the summer semester starting around 20 April 2020, and educators had about four weeks to prepare for the online semester. Due to these differences, the official pandemic measures were implemented at the start of the semester for students in Germany, while the semester was already halfway through for students from Denmark. 

Apart from trust in COVID-19 regulations, social distancing and other protective behavior is also influenced by the perceived threat of the pandemic. Responding to this kind of threat can be met by ignorance or altered behavior [22]. People often exhibit an emotional rather than rational response to a threat, also called the “rally round the flag” effect, first described by Mueller [23]. This effect is characterized by short-term support for a government during a crisis and was also evidenced in popularity polls during the COVID-19 pandemic [24]. In the long-term, the support declines again to a standard level. Particularly at the start of the COVID-19 pandemic, people were unsure about the implications for their own lives, and the fear of getting infected was shown as a predictor for personal preventive measures [25]. In the university context, the field of study and sex of students had an impact on COVID-19 risk perceptions and preventive behaviors, with female and Medicine or Health Sciences students practicing more preventive behaviors [13]. While people in both countries kept a reasonable social distance in daily life, differences were evident in the sense of social responsibility for their community and trust in public authorities, which were both higher in Denmark [26,27]. Before the official lockdown, Danish citizens had already started socially distancing and following governmental guidelines that were made official a few days after [26].

Until now, only a handful of studies have examined students’ trust in their government or university leadership during the COVID-19 crisis. Governmental recommendations were followed especially well by older students, those who felt depressed or those concerned about COVID-19 in Denmark [28]. Academic frustrations correlate with increased psychological symptoms and worries about getting infected with the virus, and psychological symptoms are negatively correlated with trust in the government [2]. Students, in particular, are not a medically vulnerable group and are not likely to experience complications due to a SARS-CoV-2 infection, but they seem to be heavily impacted by psychosocial effects [2]. It is important to know how students’ needs for psychosocial support can be addressed to help stop the spread of an infectious disease. Comparing two countries can help identify general and national-specific support and barriers to adherence. Therefore, this study has the following objectives:To describe and compare the level of COVID-19-related academic frustration, satisfaction with university regulations, risk perceptions, and trust in governmental regulations in Danish and German university students;To investigate the association between trust in governmental regulations, trust in university regulations, risk perceptions, and academic frustration among Danish and German students.

## 2. Materials and Methods

Data were collected as part of the “COVID-19 International Student Well-Being Study” (C19 ISWS), a cross-sectional online survey that was conducted in 27, mostly European, countries [29]. The study protocol and questionnaire were developed by the coordinating team of the Centre for Population, Family and Health at the University of Antwerp, Belgium (Sarah Van de Velde, Veerle Buffel, Edwin Wouters). The questionnaire is publicly accessible [30]. For the present study, the Danish and German data were analyzed. Five German universities and two Danish universities took part in the C19 ISWS. The questionnaire was designed to assess the psychological impact of the first lockdown in 2020. It was independently translated by two German members and three Danish members of the study team according to the C19 ISWS study protocol [29]. German University students were invited to complete the questionnaire in German or English and Danish students in Danish or English.

The questionnaire was released during the first lockdown in Germany and Denmark in response to the COVID-19 pandemic. In Denmark, the survey was completed between 11 May and 5 June 2020. In Germany, between 12 May and 29 May 2020 at four of the universities and between 14 July and 29 July 2020 at one university. The survey was sent to all students at the participating universities (Germany) and all students from a whole faculty (Denmark) via email distribution lists, notifications on the university webpage, social media platforms, and other websites, including e-learning platforms. A reminder email was sent out after one week, albeit only at one of the faculties in Denmark.

Any students enrolled at a university were eligible to participate in the study. The sample size aimed to reach at least 10% of all students at each university/faculty. A total of 2762 participants started the survey in Denmark (University of Copenhagen; University of Southern Denmark) and 8725 participants in Germany (Charité Berlin, University of Bremen, Heinrich Heine University Düsseldorf, Martin-Luther University Halle-Wittenberg, University Siegen). After data cleaning, the Danish sample comprised 2394 students and the German sample comprised 7506 students. The response rate in Denmark is estimated to be 10% and 18%, respectively, at the two universities. The response rate at some of the German universities where the survey link was disseminated to all students via email invitation was in the same range (10–15%). Both email and social media invitations were used at one of the German universities, making it difficult to estimate the response rate. While restrictions were partially lifted in mid-April in Denmark, this reopening phase took place in Germany at the end of April. Although in both countries, university education was online for a much longer time than this study took place.

The University of Antwerp provided a secure web platform to collect data in all participating countries (Qualtrics, Provo, UT, USA). Data protection regulations were followed in all countries, and ethical approval for conducting the study was obtained from the ethics committees at participating universities. All participants provided informed consent for taking part in the survey. An informed consent page containing the research objectives, information on data security, subjects’ privacy, confidentiality, and non-material incentive preceded the survey. Participation in the survey was voluntary, and individuals could withdraw at any time during the survey by closing the web browser. The study protocol was approved by the independent ethics committee for Social Science and Humanities at the University of Antwerp, 2020 (Case: SHW_20_38). It was also approved by the ethics committees of participating universities, while there was combined permission for the Danish universities (see Institutional Review Board Statement).

For this study, we used items assessing the sociodemographic characteristics: age, gender, educational level of parents (both parents academic, one parent academic, both parents non-academic), field of study (health-related, such as medicine or public health, vs. not health-related, such as natural sciences and humanities), and study program (Bachelor, Master, Doctor, and state exam (this degree is comparable with a diploma in specific professions, e.g., medicine)). The following scales, relevant to our study, were self-constructed based on items included in the C19 ISWS questionnaire [30]:

For assessing COVID-19-related academic frustration, we used the set of following items: (1) “I know less about what is expected of me in the different course modules/units since the COVID-19 outbreak”, (2) “I am concerned that I will not be able to successfully complete the academic year due to the COVID-19 outbreak”, (3) “The university/college provides poorer quality of education during the COVID-19 outbreak as before”, and 4) “The change in teaching methods resulting from the COVID-19 outbreak has caused me significant stress” with answering options from 1 (strongly agree) to 5 (strongly disagree), which were reversed and added to a sum score divided by the number of items. We excluded the other four items from the academic frustration score suggested by Tasso et al. [2,4] because the Cronbach alpha of the 8 item scale was insufficient (<0.65), and factor analysis has shown low factor loadings (<0.35) of the excluded items in the combined Danish/German dataset. The Cronbach alpha of the four-item scale was 0.75 in Germany and 0.77 in Denmark.

For trust in university COVID-19 regulations, the items: (1) “The university/college has sufficiently informed me about the changes that were implemented due to the COVID-19 outbreak” and (2) “I am satisfied with the way my university/college has implemented protective measures concerning the COVID-19 outbreak” with answering options from 1 (strongly agree) to 5 (strongly disagree) were reversed and added to a sum score divided by the number of items.

For trust in government COVID-19 regulations, the items: (1) “The government provided information concerning the COVID-19 outbreak on time” and (2) “The government provided comprehensive information concerning the COVID-19 outbreak” with answering options from 1 (strongly agree) to 5 (strongly disagree) were reversed and added to a sum score divided by the number of items.

For assessing COVID-19 risk perception, students were asked to rate the subjective likelihood of getting infected with SARS-CoV-2 with the question “In your opinion, how likely are you to get infected by COVID-19?” with response options from 0 (very unlikely) to 10 (very likely).

The statistical analyses were conducted with IBM SPSS Statistics 27. Descriptive statistics summarized the baseline characteristics of the study participants. Univariate analyses of variance were calculated to test country differences in the four COVID-19 related factors of interest (academic frustration, satisfaction with university regulations, risk perceptions, and trust in governmental regulations). As potential confounders, age, gender, parent education, study program, first year in higher education, and health subject were added as covariates in the models.

Finally, we evaluated in three multivariable linear models if the factors: trust in governmental COVID-19 measures, trust in university COVID-19 measures, and COVID-19 risk perception were associated with COVID-19 related academic frustration. In addition, we evaluated the same three multivariable linear models in both the Danish and the German student sample separately. To avoid collinearity, each of these factors was modeled as a potential predictor for academic frustration in a separate model. In each of the models we included age, gender, parent education, study program, first year in higher education, and health-related study subject as independent variables for adjustment.

## 3. Results

### 3.1. Baseline Characteristics

In total, 9870 participants were included after cleaning the data set (76.0% of participants were from Germany and 24.0% from Denmark). The sociodemographic characteristics of the German, Danish, and total study population are presented in Table 1. The average age was 24.6 years, with German participants, on average, two years younger than the Danish students (24.1 vs. 26.1 years). Concerning the characteristics of our sample, it is important to note that the majority of students who took part in the study (71.1%) were female, with a large number of students from health-related subjects (31.8%), especially in Denmark (62.3%). In total, 48.1% were enrolled in a Bachelor’s program, 26.9% in a Master’s program, 5.1% in a doctoral program, and 19.9% in other programs, such as the state exam in Germany. Another 22.3% of university students from Germany and 28.0% from Denmark were in their fist year of higher education. There was a difference between reported parents’ education, since in Germany, a higher percentage of one or both parents had no academic education.

### 3.2. Differences in Academic Frustration, Trust in University and Government Regulations, and Risk Perception Related to COVID-19 between Germany and Denmark

Table 2 shows differences regarding reported COVID-19-related risk perception, trust in university and governmental regulations and academic frustration between Danish and German students. With a univariate analysis of variance, we calculated differences while taking into account potential confounders such as age, gender, parental educational level, study program, and studying a health-related subject. The mean for academic frustration did not differ significantly between the countries. However, there were differences between the countries concerning trust in university COVID-19 regulations, although the means were similar (4.30 vs. 4.33). After adjustment in Denmark, students reported a significantly lower level of trust (*p* < 0.001). In contrast, trust in government COVID-19 regulations was rated higher in Denmark with a mean of 4.02, than in Germany with a mean of 3.60 (*p* < 0.001). Danish students reported a lower COVID-19 risk perception with a mean of 3.96 compared to German students with a mean of 4.23 on a scale from 1 to 10 (*p* < 0.001).

### 3.3. Associations between Trust in University Regulations, Trust in Governmental Regulations and Risk Perception, and Academic Frustration among Danish and German Students

As shown in Table 3, in the entire sample, linear regression analyses showed that trust in government as well as in university COVID-19 regulations were negatively associated with academic frustration. The strength of the association was more pronounced for trust in university COVID-19 regulations (ß = −0.372) than for trust in government COVID-19 regulations (ß = −0.125). In contrast, COVID-19 risk perception was positively associated with academic frustration.

In both the Danish and the German sample trust in government COVID-19 regulations as well as trust in university COVID-19 regulations were negatively associated with academic frustration. However, the linear regression analyses showed that risk perception was associated with academic frustration in the German sample, while no significant association was found in the Danish sample.

## 4. Discussion

This study aimed to, on the one hand, describe and compare the level of academic frustration, trust in governmental and university regulations and risk perception related to the COVID-19 pandemic. On the other hand, we investigated the association between academic frustration with risk perception, trust in governmental, and university regulations. Regarding the first research question, we found that students from both countries reported similar levels of academic frustration concerning their progress and quality of education provided by the university. Danish students showed higher trust in their government COVID-19 regulations than German students. Furthermore, higher trust in governmental and university COVID-19 regulations was associated with lower academic frustrations in both countries. 

Comparing the academic frustration across countries in our study suggests that students experienced academic frustrations at nearly the same level in both countries. A similar level of academic frustration can be explained by the need to switch to online teaching in both countries, which also explained elevated stress levels among students in other studies [4,31]. Frustration and decreased life satisfaction, specifically in students, was reported in studies from Germany, US, and UK due to isolation, lack of motivation, and concerns about learning outcomes and loneliness [14,31,32]. Academic frustration impairs students’ well-being [2], and students feel disadvantaged in the educational system because of the shift to online learning [31]. 

The finding that students’ trust in government regulations was higher in Denmark than in Germany is in line with results from other studies, suggesting that Danish citizens had greater trust in their government than in most countries, including Germany [17,20]. In addition, the perception of a timely and sufficient crisis response improved trust in government regulations [17]. A person’s trust in the government also seems to be a predictor for following governmental recommendations [33]. In general, research indicates that trust in governments is higher in citizens with higher education than among lower educated population groups in Europe, which might be explained by a higher level of social security [34]. Differences between the countries may be explained in part with a higher level of uncertainty and unclear rules in Germany. While in Germany, the federal states had different rules in place, the rules in Denmark were relatively easy to understand and uniform: For example, the maximum number of people at meetings was set to ten. It was more complicated in Germany, where rules, such as the number of people allowed to meet, depended on other factors such as being a member of one household. This pattern is also seen in the USA, where students were dissatisfied and frustrated by governmental COVID-19 regulations, which are also governed in federal states [32]. In addition to the difference in governance structure between Denmark and Germany, reporting and evaluation of COVID-19 governmental regulations in the media might have differed between the countries with consequences on the level of trust in the governmental actions in the respective populations. 

Besides a higher trust in governmental COVID-19 regulations, we found that students’ trust in university COVID-19 regulations was also higher in Denmark than in Germany. This finding is congruent with the higher trust in governmental regulations and thus mirrors general trust in public institutions. Trust in university regulations could also be affected by the point in the semester: While it was just about to start in Germany, it was already mid-semester in Denmark. Therefore, students from Denmark might be more forgiving of their institution undergoing such a sudden change. Experiences from other countries show how many factors may influence trust in university COVID-19 regulations. A survey showed that students relied on institutions like universities to protect their mental health before the pandemic in the UK. However, since being accused by politicians and media of being careless and blamed for spreading COVID-19, this perception changed [35]. Therefore, their trust in universities was reduced. At the same time, students want their universities to give out targeted advice to combat misinformation on COVID-19 among students [31].

Regarding the second research question, German students perceived a higher risk of contracting SARS-CoV-2 compared to Danish students. This might be explained by lower trust in governmental and university regulations, leading to a lower feeling of safety. In reality, the seven-day incidence per 100,000 citizens during June 2020 was comparable between both countries, between 3 and 4 in Germany and between 3 and 5 in Denmark [36,37].

Our finding that the higher the trust in governmental COVID-19 regulations, the lower the level of academic frustration among students in both countries could be explained by a lower level of stress when trust is high, as indicated by others [4,31]. In addition, we found that higher trust in university COVID-19 regulations was associated with lower levels of academic frustration both in Denmark and Germany, indicating that academic frustration could be reduced by building trust in authorities.

The association between risk perception and academic frustration differed between German and Danish students. While we found a positive association among German students, the positive association among Danish students did not reach statistical significance. This might be explained by the generally lower level of COVID-19 risk perception among Danish students and the smaller sample.

A limitation of our study is that selection bias cannot be ruled out due to the self-selection of respondents. Overall, the response rates in this study match with the expected response rates in online surveys [38]. Students might not feel inclined to answer it because it was a widespread, voluntary questionnaire. Especially when all university and most private activities were held online, such a survey could be regarded as additional screen-time. Another explanation is that particularly worried or dissatisfied students found a way to express themselves by participating in the study and could be overrepresented as a group. Another limitation is that the data was captured during the first wave of the pandemic, and therefore it is possible that opinions changed during later stages of the restrictions. Our student sample over-represents students with female identity (71.1%) and those studying health-related subjects (approximately one third), particularly in the Danish sample. This bias was somewhat reduced as we adjusted the analyses for study subject and gender. Participants’ self-reporting is another limitation in our questionnaire, especially for socially more or less acceptable items such as own academic frustration. With the cross-sectional nature of data, we were limited to associations in the data analysis. Despite these limitations, the study has important implications; it is one of the first studies to explore students’ trust in their university concerning COVID-19 regulations in a large sample size (*n* = 9870), which provides sufficient power to keep confidence intervals small.

## 5. Conclusions

Our study results underline the differences in trust in government and university COVID-19 regulations between students from Germany and Denmark. While we could show a difference between the countries concerning trust, there was no difference in their perceived academic frustration. As such, governmental and university authorities must be mindful of the needs and problems of students during this time. Further research should focus on the perceptions of students’ trust in university and governmental regulations after such a long time of measures and after the pandemic is over when COVID-19 has become endemic. It also remains unclear why there are significant differences between these culturally close neighbouring countries. While there is a difference between the countries concerning trust and fear of getting infected, there is no difference in the level of academic frustration. Despite representing a less medically vulnerable group, students’ opinions and worries should be considered. As universities continue to establish an online teaching culture, in-person social networking and teaching should still be considered an important part of students’ lives.

## Figures and Tables

**Table 1 ijerph-19-01748-t001:** Sociodemographic sample characteristics of students in seven universities in Germany and Denmark, *n* = 9870, C19 ISWS survey, 2020.

Country		Germany	Denmark	Total
		x (SD) or n; %	x (SD) or n; %	x (SD) or n; %
Participants		7506; 76.0	2364; 24.0	9870; 100
Age (years)		24.1 (4.8)	26.1 (5.9)	24.6 (5.1)
Gender				
	Female	5176; 69.0	1844; 78.0	7020; 71.1
	Male	2245; 29.9	505; 21.4	2750; 27.9
	Other	85; 1.1	15; 0.6	100; 1.0
Type of programme				
	Bachelor	3647; 48.6	1101; 46.6	4748; 48.1
	Master	1592; 21.2	1064; 45.0	2656; 26.9
	Doctor	337; 4.5	167; 7.1	504; 5.1
	Other (e.g., state exam, diploma)	1930; 25.7	32; 1.4	1962; 19.9
First year in higher education				
	Yes	1672; 22.3	663; 28.0	2335; 23.7
	No	5834; 77.7	1701; 72.0	7535; 76.3
Study subject				
	Health-related subject	1667; 22.3	1468; 62.3	3135; 31.8
	Other type of subject	5839; 77.7	896; 37.7	6735; 68.2
Parents’ educational level				
	High ^a^	2953; 39.3	1703; 72.0	4656; 47.2
	Medium ^b^	1552; 20.7	329; 13.9	1882; 19.1
	Low ^c^	2791; 37.2	288; 12.2	3078; 31.2
	Don’t know	210; 2.8	44; 1.9	254; 2.6

^a^ both parents with academic education; ^b^ one parent with academic education; ^c^ no parent with academic education.

**Table 2 ijerph-19-01748-t002:** Analysis of variance of differences in COVID-19-related academic frustration, trust in university and government regulations, and risk perception between Danish and German university students.

		Mean	SD	F	*p*-Value ^a^
Outcome: COVID-19—related academic frustration ^b^					
Denmark		3.19	0.95	3.18	0.161
Germany		3.22	0.96		
Outcome: Trust in university COVID-19 regulations ^b^					
Denmark		4.30	0.50	15.58	<0.001
Germany		4.33	0.52		
Outcome: Trust in government COVID-19 regulations ^b^					
Denmark		4.02	0.80	253.51	<0.001
Germany		3.60	0.93		
Outcome: COVID-19 risk perception ^c^					
Denmark		3.96	2.41	51.46	<0.001
Germany		4.23	2.35		

^a^ adjusted for the study program, health-related study subject, gender, age, first year in higher education and parent education; ^b^ on a scale from 1–5; ^c^ on a scale from 1–10.

**Table 3 ijerph-19-01748-t003:** Associations between COVID-19-related academic frustration (dependent variable) and trust in governmental COVID-19 regulations, trust in university COVID-19 regulations, COVID-19 risk perception in the entire, Danish and German samples. Results of seperate multiple linear regressions (*n* = 9870).

Entire Sample (*n* = 9870)	ß	SD	Standard. ß ^a^	*p*-Value	Corr. R^2^
Trust in government COVID-19 regulations	−0.130	0.011	−0.125 ^b^	<0.001	0.045
Trust in university COVID-19 regulations	−0.685	0.018	−0.372	<0.001	0.168
COVID-19 risk perception	0.011	0.004	0.028 ^c^	0.007	0.032
Danish sample (*n* = 2364)					
Trust in government COVID-19 regulations	−0.116	0.024	−0.098 ^b^	<0.001	0.064
Trust in university COVID-19 regulations	−0.672	0.037	−0.355 ^b^	<0.001	0.180
COVID-19 risk perception	0.007	0.008	0.019 ^c^	0.361	0.061
German sample (*n* = 7506)					
Trust in government COVID-19 regulations	−0.143	0.012	−0.139 ^b^	<0.001	0.045
Trust in university COVID-19 regulations	−0.685	0.020	−0.375 ^b^	<0.001	0.166
COVID-19 risk perception	0.013	0.005	0.033 ^c^	0.006	0.027

^a^ each individual model was adjusted and therefore controlled for the variables study program, health-related study subject, gender, age, first year in higher education and parent education; ^b^ coefficient per unit increment on a 5 point scale; ^c^ coefficient per unit increment on a 10 point scale.

## Data Availability

Due to the nature of this research, participants of this study did not agree for their data to be shared publicly, so supporting data are not publicly available. Data are available on request from the corresponding author for collaborating researhers within the C10 ISWS-consortium, as consent for this was provided from all participants.

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
