# Peer review of "Is Lower Trust in COVID-19 Regulations Associated with Academic Frustration? A Comparison between Danish and German University Students"

_ijerph, 2022, doi:10.3390/ijerph19031748_

Round 1
Reviewer 1 Report
[Abstract]
I could not understand why it was necessary to compare the data from Germany and Denmark after reading the abstract. Can authors explain it simply so that readers can understand it even if they only read the abstract?
I feel that the conclusion part of the abstract is not based on the results; what does it mean that there was a difference in the comparison between the two countries?
[Introduction]
The first paragraph, Line 40-41 "Most universities offered only online teaching, which precluded regular interaction with peers and lecturers"
--> It is necessary to provide some information about which region (Europe?) this description represents and when.
In this section as well, I could not understand why they dared to compare only data from Denmark and Germany.
Line 61-62 "It is still unknown if the mental health of German students worsened during the pandemic."
--> Please describe in what ways the previous studies in Reference 9-12 are inadequate. On the other hand, it is unclear whether authors consider the studies in Denmark to be similarly inadequate.
[Methods]
"COVID-19-related academic frustration", "trust in university COVID-19 regulations", "trust in government COVID-19 regulations", "COVID-19 risk perception"
-->It is not clearly stated whether these items were created by the authors on their own or based on previous studies (Reference 2 and 4?).
[Results]
Did authors find out what grade the participants is in, or how many years they have been enrolled? This could affect their trust in the university. Also, first-year students may not have much of a network with their classmates, which may increase their frustration.
Author Response
Response to Reviewer 1 Comments
[Abstract]
Point 1: I could not understand why it was necessary to compare the data from Germany and Denmark after reading the abstract. Can authors explain it simply so that readers can understand it even if they only read the abstract?
Response 1: We agree with you and give now the requested information in lines 21ff.:
“Despite the proximity of both countries, Danes and Germans differ in the level of trust in their government. This may play a role with respect to the disruptive impact of the COVID-19 pandemic on university students.”
Point 2: I feel that the conclusion part of the abstract is not based on the results; what does it mean that there was a difference in the comparison between the two countries?
Response 2: We have changed the conclusion and results in order to fit more closely to our results, lines 30ff.:
“However, German students perceived a higher risk of contracting SARS-CoV-2 compared to Danish respondents. Danish students showed higher trust in their government’s COVID-19 regulations than German students. Lower trust in government and university COVID-19 regulations and higher risk perception were associated with higher academic frustration. These results indicate that the level of trust in COVID-19 regulations might have an impact the overall frustration of students regarding their study conditions.”
[Introduction]
Point 3: The first paragraph, Line 40-41 "Most universities offered only online teaching, which precluded regular interaction with peers and lecturers"
--> It is necessary to provide some information about which region (Europe?) this description represents and when.
Response 3: We now provide the requested information in lines 41ff.:
“Most universities in Europe, including in Germany and in Denmark, offered only online teaching during the first lockdown. Online teaching limited regular interaction with peers and lecturers [3].”
Point 4: In this section as well, I could not understand why they dared to compare only data from Denmark and Germany.
Response 4: We agree that more information is needed to understand why we compared the two countries. The decision is now better justified in the introduction; lines 50ff.:
“It remains open however, whether academic frustration evolved to the same extent in different countries, because the conditions under which students live and are financed differ from country to country across Europe. Our study has a focus on Germany and Denmark, because the countries are similar with respect to demographic characteristics, extent and date of COVID-19 measures, geographical and cultural proximity [7], but at the same time there is a difference in financing due to different welfare regimes. While Danish students are not charged any tuition fees and receive a monthly allowance from the government to cover basic living costs, [8] German students have to pay an admistrative fee (€200 to €350) and only a small percentage of students receive a monthly allowance from the government (grants-and-loans-system).
Point 5: Line 61-62 "It is still unknown if the mental health of German students worsened during the pandemic."
--> Please describe in what ways the previous studies in Reference 9-12 are inadequate. On the other hand, it is unclear whether authors consider the studies in Denmark to be similarly inadequate.
Response 5: We revised the sentence to be more in line with the results of the studies; lines 69ff.:
“It is still not clear from the existing literature how the mental health of German students changed during the pandemic, while most studies indicate a deterioration in mental health for this population.”
[Methods]
Point 6: "COVID-19-related academic frustration", "trust in university COVID-19 regulations", "trust in government COVID-19 regulations", "COVID-19 risk perception"
-->It is not clearly stated whether these items were created by the authors on their own or based on previous studies (Reference 2 and 4?).
Response 6: We now provide more precise information on the origin of the items in our scales in lines 206f.:
“The following scales, relevant to our study, were self-constructed based on items included in the C19 ISWS questionnaire [30]:”
[Results]
Point 7: Did authors find out what grade the participants is in, or how many years they have been enrolled? This could affect their trust in the university. Also, first-year students may not have much of a network with their classmates, which may increase their frustration.
Response 7: Thank you very much for this relevant pont. Unfortunately, grade or years of enrollment were not measured. But it was asked, if it was students’ first year in higher education. We also included the variable “First year in higher education” in Table 1 displaying sociodemographic characteristics, the written description in lines 261f. and adjusted for it as a potential confounder in all models (see table 2 & 3).
|
First year in higher education |
|
||||
|
Yes |
1,672; 22.3 |
663; 28.0 |
2,335; 23.7 |
|
|
|
No |
5,834; 77.7 |
1,701; 72.0 |
7,535; 76.3 |
|
|
Line 262f: “22.3% of university students from Germany and 28.0% from Denmark were in their fist year of higher education.”
Reviewer 2 Report
This is an interesting study that demonstrates the impact of COVID-19 on students.
The documents are clear and well organized. Student frustration is a problem across countries. We need to care so as not to affect the future of those who support the future. This paper proposes care and prevention. Thank you for the valuable opportunity.
For readers who are not familiar with European politics, there is no clear answer as to why the two countries should be compared. I thought it would be easy to understand if there was a table comparing the situation between the two countries.
1 Introduction
Please describe the causes, mechanisms, and constituents of the frustration.
These explanations will further enhance the basis on which trust, worry, and elected.
2.Methods
The contents of the survey are written in detail.
There is a lack of analysis methods.
The stratified multiple linear regression was not specifically stated.
How were they stratified?
- Results
1)Table 3, Table4
Example for:Table3
First line: Trust in government COVID-19 regulations
y=academic frustration
X1=Trust in government COVID-19 regulations
May be age: x2,
gender: x3,…..
Is my understanding correct? If I understand correctly,
This is rarely described in the method, so you may read it as follows:
y=academic frustration
X1=Trust in government COVID-19 regulations
X2=trust in government COVID-19 regulations
X3=COVID-19 risk perception,
I also thought so at first sight. R2 made me aware of the mistake.
To avoid misunderstandings among readers, it is necessary to devise expressions such as describing them.
2)I think that if multiple regression analysis is used when comparing the two countries, it is also necessary to analyze not by country.
y=academic frustration
X1=Trust in government COVID-19 regulations
X2=trust in government COVID-19 regulations
X3=COVID-19 risk perception,
Age, gender
And Select other factors
Added country: Germany=0, Denmark:1
It may be obvious which conditions affect the most of the device.
I'm also worried about why the analysis didn't take place.
4.Discussion
1)It is repeated, "German students feel the risk of infection is high".
No reason was given. Why?
2)In order for university students to trust universities, there seems to be measures.
Is it common for young People in Europe for students to trust the government?
Could it be affected by social security richness for young people? Please mention this.
3)It may be true that student frustration has to do with low confidence in government.
Isn't it too much of a leap from the frustration of COVID-19?
Author Response
Response to Reviewer 2 Comments
Comments and Suggestions for Authors
For readers who are not familiar with European politics, there is no clear answer as to why the two countries should be compared. I thought it would be easy to understand if there was a table comparing the situation between the two countries.
Response: 1
We agree that more information is needed to understand why we campared the two countries. We did not prefer a table in the introduction , but the decision is now better justified in the introduction; lines 51ff.:
“It remains open however, whether academic frustration evolved to the same extent in different countries, because the conditions under which students live and are financed differ from country to country across Europe. Our study has a focus on Germany and Denmark, because the countries are similar with repect to demographic characteristics, extent and date of COVID-19 measures, geographical and cultural proximity [7], but at the same time there is a difference in financing due to different welfare regimes. While Danish students receive a monthly allowance from the government to cover basic living costs, and they are not charged any tuition fees (around 85% receive grants), [8] German students have to pay an admistrative fee (€200 to €350) and only a small percentage of students receive a monthly allowance from the government (grants-and-loans-system).”
1 Introduction
Point 2: Please describe the causes, mechanisms, and constituents of the frustration. These explanations will further enhance the basis on which trust, worry, and elected.
Response 2: We have included the requested description regards frustration in the introduction; lines 46ff.:
“Frustration, in general is described as a ‘blocking or prevention of a potentially rewarding or satisfying act or sequence of behaviour, or the emotional response to such hindrance’ [5, 297]. Such hindrance occurred when students had to stay at home for online teaching. Negative feelings such as disappointment, worry, fear, anxiety, tension or anger can be a consequence of this kind of interruption [6].”
2.Methods
Point 3: There is a lack of analysis methods.
The stratified multiple linear regression was not specifically stated.
How were they stratified?
Response 3: For a better understanding we revised this section in the methods part completely; lines 243ff:
“Finally, we evaluated in three multivariable linear models if the factors: trust in governmental COVID-19 measures, trust in university COVID-19 measures and COVID-19 risk perception were associated with COVID-19 related academic frustration. In addition, we evaluated the same three multivariable linear models in both the Danish and the German student sample separately. To avoid collinearity, each of these factors was modelled as a potential predictor for academic frustration in a separate model. In each of the models we included age, gender, parent education, study program, first year in higher education, and health-related study subject as independent variables for adjustment.”
Results
Point 4: 1)Table 3, Table4
Example for:Table3
First line: Trust in government COVID-19 regulations
y=academic frustration
X1=Trust in government COVID-19 regulations
May be age: x2,
gender: x3,…..
Is my understanding correct? If I understand correctly,
This is rarely described in the method, so you may read it as follows:
y=academic frustration
X1=Trust in government COVID-19 regulations
X2=trust in government COVID-19 regulations
X3=COVID-19 risk perception,
I also thought so at first sight. R2 made me aware of the mistake.
To avoid misunderstandings among readers, it is necessary to devise expressions such as describing them.
Response 4: In order to clarify that separate models were calculated for each of the independen variables, we have revised the description in the methods section (see Response 3), and we have revised the heading of Table 3, Line 304ff.
Table 3: Associations between COVID-19-related academic frustration (dependent variable) and trust in governmental COVID-19 regulations, trust in university COVID-19 regulations, COVID-19 risk perception in the entire, the Danish and German sample - Results of seperate multiple linear regressions (n=9,870)
Point 5: 2)I think that if multiple regression analysis is used when comparing the two countries, it is also necessary to analyze not by country.
y=academic frustration
X1=Trust in government COVID-19 regulations
X2=trust in government COVID-19 regulations
X3=COVID-19 risk perception,
Age, gender
And Select other factors
Added country: Germany=0, Denmark:1
It may be obvious which conditions affect the most of the device.
I'm also worried about why the analysis didn't take place.
Response 5: Thank you very much for the helpful comment. We agree with the reviewer that the clarity of our tables can be improved. We changed the wording in the tables, collapsed them into one (table 3 & 4 are now table 3) and added information in the methods section. We also included the analysis for the full sample, not stratified by country.; lines 292ff.:
“As shown in Table 3, in the entire sample, linear regression analyses showed that trust in government as well as in university COVID-19 regulations were negatively associated with academic frustration. The strength of the association was more pronounced for trust in university COVID-19 regulations () than for trust in government COVID-19 regulations (). In contrast, COVID-19 risk perception was positively associated with academic frustration.
In both the Danish and the German sample trust in government COVID-19 regulations as well as trust in university COVID-19 regulations were negatively associated with academic frustration. However, the linear regression analyses showed that risk perception was associated with academic frustration in the German sample, while no association was found in the Danish sample.”
Table 3: Associations between COVID-19-related academic frustration (dependent variable) and trust in governmental COVID-19 regulations, trust in university COVID-19 regulations, COVID-19 risk perception in the entire, Danish and German samples - Results of seperate multiple linear regressions (n=9,870)
|
Entire sample (n=9,870) |
ß |
SD |
Standard. ßa |
P-value |
Corr. R² |
|
Trust in government COVID-19 regulations |
- 0.130 |
0.011 |
- 0.125b |
<0.001 |
0,045 |
|
Trust in university COVID-19 regulations |
-0.685 |
0.018 |
-0.372 |
<0.001 |
0.168 |
|
COVID-19 risk perception |
0.011 |
0.004 |
0.028c |
0.007 |
0.032 |
|
Danish sample (n=2,364) |
|||||
|
Trust in government COVID-19 regulations |
- 0.116 |
0.024 |
- 0.098b |
<0.001 |
0.064 |
|
Trust in university COVID-19 regulations |
-0.672 |
0.037 |
-0.355b |
<0.001 |
0.180 |
|
COVID-19 risk perception |
0.007 |
0.008 |
0.019c |
0.361 |
0.061 |
|
German sample (n=7,506) |
|||||
|
Trust in government COVID-19 regulations |
- 0.143 |
0.012 |
- 0.139b |
<0.001 |
0.045 |
|
Trust in university COVID-19 regulations |
-0.685 |
0.020 |
-0.375b |
<0.001 |
0.166 |
|
COVID-19 risk perception |
0.013 |
0.005 |
0.033c |
0.006 |
0.027 |
a each individual model was adjusted and therefore controlled for the variables study program, health-related study subject, gender, age, first year in higher education and parent education; b coefficient per unit increment on a 5 point scale; c coefficient per unit increment on a 10 point scale
4.Discussion
Point 6: 1)It is repeated, "German students feel the risk of infection is high".
No reason was given. Why?
Response 6: Thank you for noticing this. We deleted the statement in the first paragraph of the discussion and now just mention it for answering the second research question; lines 367f.:
“Regarding the second research question, German students perceived a higher risk of contracting SARS-CoV-2 compared to Danish students.”
Point 7: 2)In order for university students to trust universities, there seems to be measures.
Is it common for young People in Europe for students to trust the government?
Could it be affected by social security richness for young people? Please mention this.
Response 7: Thank you for raising this point. We now include the argument in our discussion section; lines 338ff.:
“In general, research indicates that trust in governments is higher in citizens with higher education than among lower educated population groups in Europe, which might be explained by a higher level of social security [34].”
Point 8: 3)It may be true that student frustration has to do with low confidence in government.
Isn't it too much of a leap from the frustration of COVID-19?
Response 8: In this study, we only measured academic frustration related to COVID-19 conditions. Therefore, we cannot know whether general student frustration was associated with confidence in the government. We think, however, that this would be a very interesting topic for further research.
Round 2
Reviewer 1 Report
Authors have sufficiently responded to my comments. Thank you!
Reviewer 2 Report
The author has better revised the paper.
The author has made it necessary to build a relationship of trust in peacetime.